# DExD/H Box Helicases DDX24 and DDX49 Inhibit Reactivation of Kaposi’s Sarcoma Associated Herpesvirus by Interacting with Viral mRNAs

**DOI:** 10.3390/v14102083

**Published:** 2022-09-20

**Authors:** Jacquelyn C. Serfecz, Yuan Hong, Lauren A. Gay, Ritu Shekhar, Peter C. Turner, Rolf Renne

**Affiliations:** 1Department of Molecular Genetics and Microbiology, College of Medicine, University of Florida, Gainesville, FL 32610, USA; 2Genetics Institute, University of Florida, Gainesville, FL 32610, USA; 3UF Health Cancer Center, University of Florida, Gainesville, FL 32610, USA

**Keywords:** deadbox helicases, KSHV, pattern recognition receptor, innate immunity, type I interferon

## Abstract

Kaposi’s sarcoma-associated herpesvirus (KSHV) is an oncogenic gammaherpesvirus that is the causative agent of primary effusion lymphoma and Kaposi’s sarcoma. In healthy carriers, KSHV remains latent, but a compromised immune system can lead to lytic viral replication that increases the probability of tumorigenesis. RIG-I-like receptors (RLRs) are members of the DExD/H box helicase family of RNA binding proteins that recognize KSHV to stimulate the immune system and prevent reactivation from latency. To determine if other DExD/H box helicases can affect KSHV lytic reactivation, we performed a knock-down screen that revealed DHX29-dependent activities appear to support viral replication but, in contrast, DDX24 and DDX49 have antiviral activity. When DDX24 or DDX49 are overexpressed in BCBL-1 cells, transcription of all lytic viral genes and genome replication were significantly reduced. RNA immunoprecipitation of tagged DDX24 and DDX49 followed by next-generation sequencing revealed that the helicases bind to mostly immediate-early and early KSHV mRNAs. Transfection of expression plasmids of candidate KSHV transcripts, identified from RNA pull-down, demonstrated that KSHV mRNAs stimulate type I interferon (alpha/beta) production and affect the expression of multiple interferon-stimulated genes. Our findings reveal that host DExD/H box helicases DDX24 and DDX49 recognize gammaherpesvirus transcripts and convey an antiviral effect in the context of lytic reactivation.

## 1. Introduction

Human gammaherpesviruses have been implicated in numerous cancers and autoimmune disorders [1]. Kaposi’s sarcoma-associated herpesvirus (KSHV) is an oncogenic gammaherpesvirus that infects endothelial cells to give rise to Kaposi’s sarcoma (KS), and B cells to give rise to Primary Effusion Lymphoma (PEL) or Multicentric Castleman’s Disease (MCD). The seroprevalence of KSHV infection in the general population varies from <5% in the United States and Asia, to over 50% in sub-Saharan Africa, where it is estimated that 44,000 new cases of KS are diagnosed each year [2,3].

KS causes angiogenesis and dark red lesions on the skin and in the oral cavity, and leads to the transformation of endothelial cells. Although KS is most common in HIV-positive patients, other epidemiological variants occur in immune compromised individuals such as the elderly, children, or organ transplant recipients. Notably, about 1 in 200 transplant patients in the United States develop KS [4]. The disease can be life-threatening if KS reaches the gastrointestinal tract or lungs. Currently, the only treatment is the surgical removal of the lesion or chemotherapy. In contrast, PEL is a very rare and primarily HIV-associated non-Hodgkin’s Lymphoma of B cells [5]. It is a rapidly fatal body cavity-based lymphoma (BCBL) that most likely originates from post-germinal center B-cells [5]. B cells latently infected with KSHV, termed BCBL-1 cells, were able to be cultured from a PEL biopsy and have been used to study latency and reactivation [6,7].

When the immune system is weakened, individuals who are infected with KSHV are more likely to develop KSHV-related malignancies, suggesting a critical role of the immune system in suppressing cancer development [8,9,10]. The innate immune system regulates gammaherpesvirus-induced lytic reactivation in cell culture and in the gammaherpesvirus mouse model [8,11]. RIG-I and RIG-I-like receptors, members of the DExD/H box helicase family, are known to mount an innate immune response against gammaherpesviruses to inhibit viral reactivation [12,13,14]. It is becoming more evident that some of the same innate immune responses that prevent de novo KSHV infection also suppress reactivation from latency occurring within the host cell nucleus [15,16]. In the current model of KSHV infection, Pattern Recognition Receptors (PRRs) including Toll-Like Receptors (TLRs), RIG-I-Like Receptors (RLRs), and Cyclic GMP-AMP Synthase (cGAS) can recognize KSHV and other herpesviruses to stimulate an innate immune response [12,16]. During KSHV reactivation, knockdown of RLRs led to increased lytic reactivation in HEK-293 cells [13]. There are multiple RNA regions in KSHV that could act as potential virus ligands of RIG-I to induce type I interferon responses [17]. However, the mechanism by which RIG-I senses KSHV from within the nucleus during reactivation is unknown. It is uncertain whether RLRs directly recognize viral dsRNA since cGAS, an antiviral DNA-sensing molecule, also recognizes KSHV dsDNA and is involved in crosstalk with RIG-I to stimulate interferon (IFN) genes [18,19].

RNA helicases recognize and bind to segments of double stranded RNA and are involved in many cellular processes including RNA transport, localization, splicing, and translation [20]. The helicases recognize their RNA targets by a combination of secondary structure and by interacting with other RNA binding protein partners [21,22]. When bound, RNA helicases displace RNA:protein complexes to facilitate RNA processing and, in some cases, helicases also have the ability to translocate, and unwind RNA:RNA duplexes [23]. During viral infection, a subset of RNA helicases has the ability to recognize non-self dsRNAs via location in the cell, structural defects such as bulges or loops, and/or nucleotide modifications that all can lead to stimulation of an innate immune response [24,25,26,27].

RIG-I (DDX58) and RIG-I-like receptors are members of the largest superfamily of RNA helicases, termed DExD/H box helicases. This family of helicases are RNA-binding proteins that utilize ATP to unwind short duplex regions of RNA and remodel RNA-protein complexes, thereby altering RNA secondary structure. They are characterized by the DExD/H box helicase domain, a conserved ATP binding motif that contains the amino acid sequence Asp-Glu-x-Asp/His [28,29]. The core homologous DExD/H box domains are centered in the protein and a helicase domain is often at the C-terminal end [29]. Disordered flanking segments at the C-terminal and N-terminal ends are up to hundreds of amino acids in length and are thought to contain linker regions involved in interactions with other proteins or RNAs [30].

There are about 60 DExD/H box helicases in mammalian cells [31]. Based on crystal structures of DExD/H box proteins complexed with RNA, the binding sites in the helicase core contact the RNA backbone exclusively, allowing association with duplex RNA but excluding preferential binding of particular RNA sequence motifs [32]. When a cell is infected with a virus, several DExD/H box helicases that are not part of the RIG-I family have recently been shown to act as viral PRRs or signaling adaptors of innate immune pathways. Some examples are DDX3, DDX41, and DDX24 [30,33,34,35,36,37].

To identify if other DExD/H box helicases can act as PRRs or adaptors of the innate immune system during KSHV viral reactivation, we performed an siRNA knockdown screen of candidate helicases, and identified DDX24 and DDX49 as inhibitors of KSHV lytic reactivation. DDX24 and DDX49 immunoprecipitation followed by RNA enrichment and high-throughput sequencing analysis of helicase-bound RNAs yielded numerous KSHV immediate-early (IE) and early (E) transcripts. Ectopic expression of IE and E KSHV transcripts in the absence of viral infection induced type I interferon production. Moreover, the interaction of DDX24 and DDX49 with viral IE and E transcripts during reactivation stimulated multiple interferon-regulated genes.

## 2. Materials and Methods

### 2.1. Cell Culture

Vero rKSHV.219 cells [38], a gracious gift from the Michael Lagunoff lab, were maintained in DMEM with 10% FBS and 10 μg/mL puromycin and induced with 20 ng/mL tetradecanoyl phorbol acetate (TPA) and 2 mM sodium butyrate (NaB). Knockdown BCBL-1 cells were maintained in RPMI 1640 with 10% FBS and 1 μg/mL puromycin, and induced with 20 ng/mL TPA and 2 mM NaB. Electroporated BCBL-1 cells expressing DDK-tagged DExD/H box helicases were maintained in RPMI 1640/10% FBS with 500 μg/mL G418 and induced with 2 mM NaB. TREx BCBL1-Rta cells were engineered by the Jae Jung lab using Flp-mediated site-specific recombination to be inducible for Rta following doxycycline treatment [39]. Lentiviral transduced DExD/H box-Avi overexpressing TREx BCBL1-Rta cells were maintained in RPMI 1640/10% FBS with 1 μg/mL puromycin and induced with 1 μg/mL doxycycline and 0.25 mM phosphonoformic acid (PFA).

Cells of THP-1, an acute monocytic leukemia cell line, were incubated in 5% CO_2_ at 37 °C using RPMI 1640 Medium with 10% fetal bovine serum, 2 nM L-glutamine adjusted to contain 1.5 g/L sodium bicarbonate, 4.5 g/L glucose, 10 mM HEPES and 1.0 mM sodium pyruvate and supplemented with 0.05 mM 2-mercaptoethenol. Cells were maintained at 10^5^ to 10^6^ cells/mL. HEK-blue IFNα/β™ cells were cultured in 5% CO_2_ at 37 °C using DMEM medium with 10% FBS. Cells are selected with 30 µg/mL blasticidin and 100 µg/mL Zeocin. Cells were subcultured when they reached 70~80% confluency.

### 2.2. Helicase siRNA Screen

Vero rKSHV.219 cells were seeded in individual wells of a 48-well plate at 200 μL of 9.5 × 10^4^ cells/mL in DMEM/10% FBS without antibiotics and grown overnight. A Lipofectamine RNAiMAX transfection (Invitrogen, Thermo Fisher Scientific, Waltham, MA, USA) was performed with a pool of 5 ON-TARGETplus siRNA duplexes that target a particular DExD/H box helicase or scrambled control (Dharmacon, Horizon Discovery, Lafayette, CO, USA) for a final siRNA concentration of 50 nM as per the manufacturer’s instructions. The list of the 22 DExD/H box helicase targets used in the knockdown screen [40] is presented in Appendix A. The sequences of siRNAs against specific helicases and controls are in Appendix A. After 28 h, the knock-down of control RNAi GFP was confirmed and all siRNA containing media was replaced with 10% FBS media containing 20 ng/mL TPA, and 2 mM NaB to induce lytic reactivation. Fluorescence images were acquired on a Leica DMI 4000 microscope, cell counts were recorded, and DNA was extracted at 48 h post-induction. Three random fields of vision were selected to quantify the adherent Vero cells prior to the 29 h post-induction time point. After the 36 h time point, lytic Vero cells were no longer adherent and could not be quantified.

### 2.3. Real-Time qPCR

For RT-qPCR, RNA was extracted from KSHV infected cells throughout the induction time course using RNA Bee (Tel-Test, Friendswood, TX, USA). Total RNA was quantified using Nanodrop 1000 spectrophotometer (Thermo Fisher Scientific, Waltham, MA, USA), and 0.9 μg Vero RNA or 1.6 μg BCBL-1 RNA was Turbo DNase treated (Invitrogen, Carlsbad, CA, USA), then converted to cDNA with the High-Capacity RNA-to-cDNA Kit (Thermo Fisher Scientific, Waltham, MA, USA). All cDNA analyses were quantified relative to GAPDH.

DNA was extracted using a QIAamp DNA Mini kit (QIAGEN, Valencia, CA, USA). KSHV genome copy numbers from 30 ng of total DNA were quantified utilizing LANA N-terminus primers based on a standard curve constructed from known amounts of pcDNA3.1 plasmid DNA. Quantitative, real-time PCR (qPCR) was performed on an ABI StepOnePlus (Applied Biosystems, Foster City, CA, USA) for the Vero intracellular cDNA/DNA and a Roche Light Cycler 96 (Roche, Indianapolis, IN, USA) for the BCBL-1 intracellular cDNA/DNA. DNase-treated RNA was converted to cDNA using RNA-Quant^TM^ cDNA synthesis kit (System Biosciences, Palo Alto, CA, USA) to include RNA classes with strong secondary interactions. Primer sequences are listed in Appendix A.

### 2.4. Transfection by Electroporation for BCBL-1 Overexpression

1 × 10^7^ BCBL-1 cells were grown in antibiotic-free RPMI. The next day, they were pelleted and rinsed twice in PBS. The cell suspension was transferred into a 0.4 cm electrode gap cuvette with 10 μg of TrueORF Gold pCMV6-Entry DNA vector (Origene, Rockville, MD, USA) containing the Myc-DDK (FLAG)-tagged helicase of interest (Origene, Rockville, MD, USA) or pmaxGFP (AddGene, Cambridge, MA, USA). Cuvettes were treated 2 times at 250 V, 950 μF with the Gene Pulser Xcell device (Bio-Rad, Hercules, CA, USA). Electroporated BCBL-1 cells were grown for 48 h in non-selective RPMI medium. After the 48 h, media was replaced with RPMI containing 400 μg/mL G418 for 4 weeks.

### 2.5. Antibodies and Western Blot Analysis

Monoclonal mouse antibody against DDK (TA50011) was from Origene (Rockville, MD, USA). Monoclonal mouse antibody against Avi was from Genscript (Piscataway, NJ, USA). Goat αDHX29 (sc-107197) and goat αGAPDH (V-18, sc-20357) were from Santa Cruz (Santa Cruz, CA, USA). Mouse monoclonal α-Tubulin (CP06-100UG) was from Oncogene/EMD Millipore (Burlington, MA, USA). IRDye 800-labeled IgG and IRDye 680-labeled IgG Donkey α-goat (925-68074) and α-mouse (925-32212) secondary antibodies were from Li-Cor Biosciences (Lincoln, NE, USA).

8 × 10^5^ cells were pelleted by centrifugation, washed in cold PBS, and resuspended in 80 μL RIPA buffer containing 150 mM NaCl, 50 mM Tris (pH 8), 1% (*v*/*v*) NP-40, 0.5% sodium deoxycholate, 2 mM EDTA, and 0.1% SDS with protease inhibitor cocktail (Roche, Indianapolis, IN). Pierce BCA protein assays were performed (Thermo Fisher Scientific, Waltham, MA, USA). Then, 20–40 μg aliquots (25 μg for overexpressing BCBL-1 cells, 20 μg for DHX29 KD, and 40 μg for TREx BCBL1-Rtas) of total protein were separated by 10% SDS-PAGE gels and transferred to a nitrocellulose membrane (Bio-Rad, Hercules, CA, USA). The membranes were blocked with Odyssey blocking buffer (TBS) from Li-Cor Biosciences (Lincoln, NE, USA) at 4 °C overnight and then subsequently incubated with primary antibodies (1:2000 α-DDK, 1:200 α-GAPDH for overexpressing BCBL-1 cells; 1:200 α-DHX29, 1:1000 α-tubulin for DHX29 KD; 1:1000 α-Avi, 1:1000 α-GAPDH for TREx BCBL1-Rtas) for 1 h and secondary antibodies (1:15,000) for 1 h at room temperature. After washing, the membranes were scanned with an Odyssey CLx infrared imaging system (Li-Cor, Lincoln, NE, USA) at wavelengths of 700 and 800 nm to quantify protein bands, and the molecular sizes of the developed proteins were determined by comparison with pre-stained protein markers (Bio-Rad, Hercules, CA, USA).

### 2.6. Lentiviral Stable Transduction for BCBL-1 Knockdown

Hairpin shRNAs in the pLKO.1 lentiviral vector designed by The RNAi Consortium (TRC) targeting DDX24, DDX49, DHX29, and DDX58 DExD/H box helicases were cloned (Dharmacon, Horizon Discovery, Lafayette, CO, USA). Then, 5 μg of each shRNA vector, 2.5 μg psPAX2, and 2.5 μg pMD2.G were co-transfected with TransIT-293 Transfection reagent (Mirus Bio, Madison, WI, USA) in 9 cm plates containing 5 × 10^6^ HEK 293FT cells, as per the manufacturer’s protocol. Eighteen hours after transfection, plasmid-containing media was replaced with fresh DMEM. Seventy-two hours post transfection, the supernatants were collected. All lysates were centrifuged at 3000 rpm for 5 min, 0.45 μm filtered, and stored at −80 °C prior to infection. Then, 5.0 × 10^6^ BCBL-1 cells were resuspended in a 5 mL pool of pure viral supernatant, each containing 1 mL of lentiviral stock targeting a specific DDX or a control lentivirus with empty pLKO.1 construct for 24 h. A pool of 5 different shRNA expressing lentiviruses were used for knock down (Appendix A). All lentivirus infected cells were selected in RPMI with 10% FBS and 1 μg/mL puromycin for 13 days and frozen in liquid nitrogen for future analysis.

### 2.7. Lentiviral Transduction for TREx BCBL1-Rta Overexpression

Lentiviral pLV constructs expressing Avi-FLAG-tagged versions of maxGFP, DHX29, DDX24, DDX49, and DDX58 (designated by NCBI accession number) were obtained from VectorBuilder (Santa Clara, CA, USA). TREx BCBL1-Rta cells were grown under 200 μg/mL hygromycin B selection to a density of 5 × 10^5^ cells/mL. A total of 2 × 10^6^ TREx BCBL1-Rta cells were infected with lentivirus at an MOI of 10 with 8 μg/mL polybrene. Twenty-four hours post infection, the virus containing media was replaced by media containing 0.25 μg/mL puromycin RPMI and cells were grown under selection for 7 days. Then, the puromycin concentration was gradually increased to 1 μg/mL for another 5 days. GFP expressing cells were monitored periodically for successful expression of the transgene. The infection was performed twice independently (two biological replicates).

### 2.8. RNA Immunoprecipitation

TREx BCBL1-Rta cells overexpressing maxGFP, DDX24, DDX49, or DDX58 were grown to a density of 5 × 10^5^ cells/mL in 100 mL. One set of flasks was harvested during latency, the other was reactivated in RPMI media containing 1 μg/mL Doxycycline and 0.25 mM PFA for 24 h. When cells were harvested, they were incubated in lysis buffer (20 mM MOPS-KOH pH 7.4, 120 mM KCl, 0.5% Igepal, 2 mM β-Mercaptoethanol supplemented with 200 unit/mL RNasin (Promega) and Complete Protease Inhibitor Cocktail (Roche)) for 20 min on ice. The lysate was cleared by centrifugation and endogenous proteins were immunoprecipitated overnight at 4 °C with 10 μg total of α-Avi tag monoclonal mouse antibody, (2 µg/ul, Genscript, catalog# A01738) composed of a mouse IgG2a heavy chain. Then, 5 mL of the antibody-bound RNA lysate was incubated with 200 μL of protein G MagBeads MX (Genscript, Piscataway, NJ) for 2 h at 4 °C. The beads were washed three times with MOPS buffer and eluted with 0.1 M glycine, pH 2.2 as per the protocol. The RNA was isolated by TRIzol reagent for RNA extraction (Sigma-Aldrich, St. Louis, MO, USA), TURBO Dnase treated (Invitrogen), and subjected to further analysis. Input and IP samples from non-transduced, DDX24-Avi, and DDX49-Avi TREx-BCBL1-Rta cells were collected for sequencing.

### 2.9. Next Generation Sequencing (NGS)

Libraries were prepared using the TruSeq Stranded Total RNA kit with Ribo-Zero Gold (Illumina), and were tested on TapeStation (Agilent) to verify the cDNA quality and size for sequencing. Sequencing was performed by the Interdisciplinary Center for Biotechnology Research (ICBR) at the University of Florida on the Illumina HiSeq3000 platform to generate 50 million, 100 base paired-end reads for each sample.

The sequencing data were processed and analyzed as follows. First, Illumina adapter sequences were removed from the reads using Trimmomatic. The quality of the reads was then verified with FastQC. After these pre-processing steps, a paired-end alignment was performed against the KSHV BAC16 genome (GQ994935.1) using Bowtie2. The number of aligned reads was normalized using Counts per Million (CPM) for each sample. The CPM values were used to calculate coverage on either strand of KSHV genome. The read coverage of the control samples, which represented non-specific background, was subtracted from those of the experimental DDX24 and DDX49 samples. The normalized coverage tracks of DDX24 and DDX49 were then visualized using Integrative Genomics Viewer tool.

### 2.10. Statistical Analyses

For all experiments, mean ± standard error mean (S.E.M.) were calculated and significance was determined by performing unpaired two-sample Student’s *t*-tests, using GraphPad Prism software (GraphPad Software, La Jolla, CA, USA). ANOVA was used for grouped analyses.

### 2.11. Type I Interferon Detection by SEAP Assay and Transcript Measurement by RT-qPCR

5 × 10^5^ THP-1 cells were transfected with 5 µg poly(I:C) (InvivoGen), or with 5 µg pcDNA3.2 plasmids containing different KSHV genes, GFP, LANA and empty pcDNA3.1 via Lipofectamine 3000 transfection kit (Thermo Fisher Scientific). The KSHV genes included K2, ORF70, K4, K4.1, K4.2, K5, K9, ORF58, ORF59, ORF65, ORF66, and ORF 67. The mammalian expression vector plasmids expressing KSHV transcripts were gifts from Dr. Denise Whitby in Frederick National Laboratory for Cancer Research (listed in Appendix A). The pcDNA3.2 derivatives have the CMV promoter, neo and amp resistance. Supernatants were harvested at 12 h and tested for type I IFN activity using HEK-blue IFNα/β™ cells, which were engineered to monitor JAK-STAT pathway activation by expression of a secreted embryonic alkaline phosphatase (SEAP). SEAP activity produced in response to IFNα/β was measured with a chromogenic substrate.

A total of 50,000 HEK-blue IFNα/β™ cells were seeded in 180 µL QUANTI-Blue medium in each well of a 96-well plate. Supernatant from THP-1 cell culture medium (20 µL) was added per well. IFN-α and IFN-β were assessed by measuring SEAP activity using optical density (OD) at 620–655 nm with a microplate reader. Two biological replicates and three technical replicates were performed in each experiment.

RNA was extracted from THP-1 cells 12 h post transfection using TRIzol (Invitrogen) and quantified. RNA was converted to cDNA using the high-capacity RNA to cDNA kit (Thermo Fisher Scientific) in a total volume of 20 μL. Real-time PCR was performed using primers targeting human IFN-α and IFN-β, with GAPDH as an endogenous control.

## 3. Results

### 3.1. Knockdown of DexD/H Box Helicases to Screen for Effects on KSHV Reactivation

To determine if other DExD/H box helicases in addition to known RLRs affect KSHV lytic reactivation, we performed a transient siRNA knock-down (KD) screen of 22 individual DExD/H box helicases (Appendix A) that previously found to have an effect on Myxoma virus infection efficiency [40]. Five siRNAs were used to potently knockdown each DExD/H box helicase to ensure a knockdown as well as patterned modifications for a low off-target effects. The Vero cell line is latently infected with a recombinant virus, rKSHV.219 which expresses the red fluorescent protein (RFP) from the KSHV early lytic PAN promoter, and the green fluorescent protein (GFP) from the EF-1α promoter, and with the gene for puromycin resistance as a selectable marker. Therefore, it facilitates the detection of latent and immediate-early lytic stages of infection via fluorescent markers [38]. Following transfection with siRNA pools targeting individual helicases, the cells were induced at 29 h with tetradecanoyl phorbol acetate (TPA) and sodium butyrate (NaB).

Within 24 h, the DHX29 knockdown demonstrated a 90% reduction in reactivation as measured by the percentage of RFP-expressing cell number, and notably, two of the screened helicases, DDX49 and DDX24, showed a 2-fold increase in reactivation (Figure 1A,B). No significant difference was observed for the remaining 19 DExD/H box helicases (Appendix A). The sequences of the pooled siRNAs used to knockdown DHX29, DDX24, and DDX49 and for the non-targeting control pool are shown in Appendix A. It should be noted that all knockdowns without induction showed negligible background reactivation. Additionally, all cells appeared viable until the chemical inducers were added to the media to activate the KSHV lytic cycle. When RIG-I alone was knocked down in Vero rKSHV.219 cells during the screen, there was no observable change in reactivation when compared to the non-specific control.

To confirm that RFP expression truly represented lytic DNA replication, intracellular KSHV genomic DNA was measured by qPCR with primers that amplify the C-terminus of ORF73 encoding LANA. Primer sequences are listed in Appendix A. qPCR results estimate the viral copy number at 48 h post-induction representing active KSHV replication. Relative to knockdown with a scrambled siRNA control, DHX29 knockdown had about 2/5 the number of genome copies, DDX49 knockdown demonstrated a 1.5-fold increase, and DDX24 knockdown demonstrated a 3-fold increase in the number of viral genome copies (Figure 1C). Hence, supports that DHX29 acts as a pro-viral sensor, whereas DDX24 and DDX49 act as antiviral sensors, which is in congruence with our observations from fluorescence microscopy.

Although Vero rKSHV.219 offered a simple tool for a reactivation screen, Vero cells do not produce type I interferons, one of the two downstream pathways of MAVS signaling. Thus, IFN-competent human BCBL-1 cells were used for further phenotypic studies.

To monitor the effects on KSHV reactivation, a DHX29 siRNA expressing lentivirus was prepared via second generation packaging of the pLKO.1 plasmid to achieve a stable knockdown in BCBL-1 cells. qPCR confirmed a 50% decrease in DHX29 mRNA levels (Appendix A). However, we observed no significant KSHV gene expression changes upon viral reactivation in the DHX29 KD cells. One explanation for this is the above-mentioned missing Type I interferon production in Vero cells. Therefore, the remainder of this study focuses on the downstream analysis of the potentially antiviral PRRs, DDX24 and DDX49.

### 3.2. Construction of Stably Transfected BCBL-1 Cells Expressing DDX24-DDK and DDX49-DDK

Stable transformants with increased levels of DDX24 and DDX49 were constructed. BCBL-1 cells were transfected via electroporation with plasmid constructs containing DDK-tagged versions of complete DEAD-box open reading frames (Figure 2). Primers targeting the gene body of DDX24 and DDX49 demonstrated an increase in their expression in stably transfected DDX24-DDK and DDX49-DDK cells, respectively, compared to the controls with empty pCMV6 vector, which contained only endogenous DDX24 and DDX49 (Figure 2A). Despite a relatively high abundance of endogenous DDX24 in BCBL-1 cells (based on normalization to GAPDH mRNA), an increase in expression was still observed after ectopic overexpression. Additionally, primers exclusively targeting the DDK tagged 3′ end of the cDNA confirmed the overexpression of exogenous DDX24-DDK and DDX49-DDK in transduced cells (Figure 2B). An immunoblot utilizing an anti-DDK antibody confirmed protein expression of the transgenic helicases, DDX24-DDK and DDX49-DDK (Figure 2C). The survival assay for DDX24-DDK and DDX49-DDK stably transfected BCBL-1 cells during the induction time course (Figure 2D) demonstrated that population cell viability was unaffected in the stably transduced BCBL-1 cells and overexpression of either DDX24 or DDX49 did not determine KSHV viral production in this experiment.

### 3.3. Effects of DDX24-DDK and DDX49-DDK on KSHV Gene Expression following Induction

Next, viral latent, immediate-early, and late lytic gene expression were monitored in a time course experiment for up to 72 h post-induction by RT-qPCR (Figure 3). Primer sequences are listed in Appendix A. Upon induction, the transcription of the latency antigen, LANA was significantly decreased in the DDX24-DDK and DDX49-DDK overexpressing (OE) cells (Figure 3A,D). The expression of immediate-early gene RTA, the master regulator of lytic KSHV gene expression, was strongly inhibited throughout the 72 h time course upon DDX24 or DDX49 overexpression (Figure 3B,E). Inhibition of RTA suggests that the effects of these host helicases occur very early during reactivation. Late lytic, K8.1, gene expression also remained inhibited throughout the entire induction time course in the DDX49 overexpressing cells (Figure 3C,F).

Additionally, to monitor active replication, qPCR was performed on intracellular viral DNA over a time course of 72 h. By 48 h, both DDX24-DDK and DDX49-DDK overexpressing BCBL-1 cells had about 2/3 the number of viral genome copies of that observed in control transfected cells (Figure 4). Hence, in addition to lytic gene expression, KSHV genome replication was also reduced in these stably transfected cells. This further demonstrates that reactivation was suppressed when DDX24 or DDX49 was overexpressed.

### 3.4. Stably Transduced TREx BCBL1-Rta Cells Expressing DDX24-Avi and DDX49-Avi

Based on the knockdown and overexpression data, there appears to be a functional role of DDX24 and DDX49 in inhibiting KSHV reactivation. We demonstrated a phenotype and wanted to characterize the biological role of these helicases during latent and lytic KSHV infection. It is well established that RIG-I interacts with RNAs during viral infection, and therefore we asked whether DDX24 and DDX49 also directly interact with viral RNAs [41,42]. DDX24 and DDX49-specific RNA immunoprecipitation (RIP) was used to identify the viral RNAs that bind to these helicases. RNA-IP experiments were performed in TREx BCBL1-Rta cells [39] over expressing DDX24 or DDX29. TREx-BCBL1-Rta cells are BCBL-1 cells that contain the lytic Rta transcriptional regulator of KSHV downstream of a tetracycline inducible promoter. In this system we can achieve a more robust induction. Between 60–80% of the total TREx BCBL1-Rta cell population will undergo reactivation, compared to only about 10–20% in induced BCBL-1 cells.

To achieve high specificity and expression, we used Avi/FLAG-tagged DexD/H box helicase expressing lentiviruses. The Avi fusion protein contains a 15 amino acid Avi tag (GLNDIFEAQKIEWHE) at the C-terminal end, which allows highly specific IP without contamination of other highly conserved cellular helicases [43]. Three biological groups of TREx-BCBL1-Rta cells were transduced: DDX24, DDX49 and plus GFP, which was used as expression control.

Western blot and qPCR were used to assess the expression of the Avi-tagged helicases in TREx BCBL1-Rta cells. The DDX24-Avi and DDX49-Avi recombinant cells were both able to express the helicase of interest (Figure 5).

### 3.5. RNA Immunoprecipitation of TREx BCBL1-Rta Cells

To identify potential viral RNAs that are bound by tagged DDX24 and DDX49 through next generation sequencing, TREx BCBL1-Rta cells, either transduced by maxGFP or overexpressing DDX24 or DDX49, were grown and harvested either during latency or at different time-points post induction with doxycycline. To verify that the TREx BCBL1-Rta cells were efficiently induced under these conditions, the expression of lytic viral RTA and ORF59 genes was measured by RT-qPCR (Appendix A). As evidenced by the fold change of RTA and ORF59 after induction, TREx BCBL1-Rta cells were robustly induced.

Cell lysates were immunoprecipitated overnight with α-Avi antibody and the antibody-bound lysate was then incubated with beads bound to the heavy chain. RNA was extracted by TRIzol (Sigma-Aldrich), treated with Dnase (Invitrogen), and was then used to prepare libraries for sequencing. To ensure that immunoprecipitation enriched for the Avi-tagged helicases, Western blot assay was performed to detect α-Avi before and after streptavidin bead selection (Appendix A). After the concentration of the immunoprecipitated lysate, a single distinct band was visible for DDX49-Avi at the correct size, 54 kDa, indicating that Avi-tagged proteins are selectively immunoprecipitated.

The reads obtained from sequencing of avi-tag pulled-down RNA were aligned to the KSHV reference genome which showed high coverage at specific KSHV loci especially for immediately early and early gene transcripts (IE, E) (Figure 6), strongly suggesting that DDX24 and DDX49 inhibit KSHV reactivation from within the nucleus by directly interacting with KSHV mRNAs.

Since RIP analysis was conducted under non-crosslinking conditions, together with stringent wash steps, only high energy interactions between the helicase and bound RNAs should be recovered. There were almost no reads from maxGFP control wildtype cells compared to the high number of reads resulting from the DDX24 and DDX49 overexpressing cells, which supports the specificity of the RIP experiment. As expected, we observed much lower reads in the latent samples compared to the lytic samples. The DDX24 and DDX49 helicases were observed to bind to several identical KSHV RNAs during lytic infection with varying degrees (Figure 6). We also observed divergent RNA interactions between DDX24 and DDX49, indicating some degree of binding specificity between these two helicases. As previously mentioned, DexD/H box proteins interact with the RNA backbones in a mostly sequence non-specific manner [21]. The unique recognition and downstream enzymatic functions of each of the helicases in vivo are most likely determined by their protein binding cofactors. Three KSHV transcripts, K9, ORF37/38 and ORF 65/66/67 (Figure 6), were found enriched in the DDX49 pulldown but did not associate with DDX24.

There were five regions that had the greatest transcript enrichment for both helicases: K2/ORF70/K4/K4.1/K4.2, K5, K7, K8/K8.1 and the ORF58/59 region (Table 1, Figure 6). All these genomic loci include genes that encode for IE and E proteins and many of which are involved in immune modulation, or reactivation. K2 is an IE gene that encodes for vIL-6, which activates JAK/STAT, Mitogen Activated Protein Kinase (MAPK), and Akt signaling pathways to regulate B-cell proliferation, KSHV reactivation and is crucial for B cell survival during PEL [44]. K5 encodes for a ubiquitin ligase that strongly downregulates MHC class I from the cell surface thereby contributing to the immune evasion [45].

### 3.6. Testing of Cloned KSHV Genes for Their Ability to Induce a Type I Interferon Response

It is well-known that during de novo infection, detection of cytosolic KSHV DNA leads to activation of type I interferon production. However, fewer studies have focused on the PRR-dependent recognition of viral RNAs in the context of reactivation of latent viruses from the host cell nucleus. To investigate whether KSHV transcripts generated within the cells during viral reactivation can also affect host innate immune response, THP-1 cells derived from an acute monocytic leukemia were transfected with plasmids encoding candidate IE and E KSHV genes identified from our RIPseq dataset (Figure 6), and type I interferon (IFN) production was assessed. A synthetic dsRNA, Polyinosine-polycytidylic acid (poly(I:C)), was used as a positive control. Poly(I:C) is a molecular pattern associated with viral infection. Poly(I:C) activates the antiviral pattern recognition receptors TLR3, RIG-I/MDA5 and PKR, thereby inducing signaling via multiple inflammatory pathways, including NF-kB and IRFs. A time course was performed to measure type I IFN produced in response to poly(I:C) and 12 h was chosen as the optimal harvest time to examine IFN response to KSHV transcripts (Appendix A). pcDNA3.1 plasmids containing 12 different KSHV genes (Appendix A) whose transcripts were identified as being bound by DDX24 and DDX49, were transfected into THP-1 cells. LANA (not enriched during RIPseq), GFP, and empty vector (pcDNA3.1) were used as negative controls. HEK-blue™ IFNα/β cells were utilized to measure secreted Interferon alpha (IFNα) and Interferon beta (IFNβ). HEK-blue™ IFNα/β cells contain a secreted embryonic alkaline phosphatase (SEAP) reporter gene expressed from a type I interferon-stimulated promoter.

Poly(I:C) gave a strong signal, and none of the negative controls (empty vector, GFP or LANA) resulted in significant IFN production (Figure 7A). In contrast, all 12 plasmids containing KSHV sequences (K2, ORF70, K4, K4.1, K4.2, K5, K9, ORF58, ORF59, ORF65, ORF66, and ORF67) were able to stimulate secreted type I interferon protein production at levels above the negative controls (Figure 7A). These results are consistent with specific KSHV transcripts triggering IFNα/β production partially through binding to DDX24 and DDX49.

To investigate whether the increase in type I IFN production was accompanied by an increase in the steady-state levels of IFN transcripts, RT-qPCR analysis was performed in transfected THP-1 cells. Since IFN-α has a very short half-life, about 2–3 h [52], results for IFN-β transcripts are shown in Figure 7B. IFN-β mRNA expression was highly increased in response to the positive control poly(I:C). In contrast, no significant induction of IFN-β mRNA was observed in cells transfected with the plasmids encoding KSHV IE and E genes that induced type I IFN protein (Figure 7A). Therefore, these results suggest that KSHV IE and E mRNAs stimulate type I interferon production via a post-transcriptional mechanism upon KSHV reactivation.

## 4. Discussion

The biomedical significance of RLRs has been rising rapidly due to their close relatedness to cancer development and progression. They elicit an interferon-mediated response upon recognition of specific nucleic acids derived from viral pathogens. Our study has revealed another two RLRs, DDX24 and DDX49, that function to recognize viral nucleic acids, in addition to well-studied RIG-I and MDA5.

The siRNA screen in Vero cells revealed an antiviral activity for DDX24 and DDX49. Previous studies reported cellular helicases that are required for viral life cycles [53], for example, it has been shown that the host DDX3 protein is required for HIV mRNA Rev-RRE export function [54,55]; while the cell also senses abortive HIV-1 transcription products to induce a type I interferon response [55].

DDX24 predominantly localizes in the nucleus [56] and is known to interact with both the innate immune system and KSHV. The DDX24 promoter region has binding sites for several interferon-regulated transcription factors such as STAT1 and IRF7. Additionally, DDX24 interacts with USP7 in the DEAD-box helicase core region (EGPS sequence) and preferentially impedes recruitment of IRF7 by associating with the adaptor proteins FADD and RIP-1 of the caspase pathway [37,57]. Additionally, microRNAs from several oncogenic herpesviruses have recently been reported to target DDX24. quick Crosslinking and Sequencing of Hybrids (qCLASH) datasets demonstrated that KSHV [58], Murine gammaherpesvirus 68 (MHV68) [59] and EBV microRNAs [60] all target DDX24 and negatively regulate DDX24 expression.

DDX49 helicase is localized in the nucleus [61] and is required for regulating RNA transcription, stability, and efficient export of non-spliced poly (A)+ RNAs from the nucleus [61]. Interestingly, the majority of KSHV lytic transcripts are not spliced and require nuclear export facilitated by ORF57 during lytic [62]. Moreover, KSHV also encodes ORF10 which inhibits host cellular spliced mRNA export from the nucleus [63]. This further indicates that unlike most of the other PRRs, DDX49-dependent recognition process occurs within the nucleus, where KSHV reactivation initiates. DDX49 may also play a role in oncogenesis since its up-regulation has been reported for multiple malignancies [61].

Vero cells are IFN deficient, suggesting that DDX24 and DDX49 may be acting via an alternate signaling path to suppress reactivation. Based on current literature, the IRF7 and IRF3 pathways may also be involved in DDX24 signaling [37].

To investigate the effects of DDX24 and DDX49 on viral replication, we generated BCBL-1 cells stably overexpressing DDX24-DDK or DDX49-DDK. Over-expression of DDX24 or DDX49 in BCBL-1 cells inhibited KSHV genome replication and expression of immediate-early genes, suggesting that these helicases act during early stages of reactivation. In our study, DDX24 and DDX49 are found to interact with and recognize IE and E transcripts at the beginning of reactivation. Since both helicases are located in the nucleus, where early lytic intron-less KSHV mRNAs should locate for mRNA export, it would be interesting to test whether there is a synergistic effect of both helicases [62].

Multiple studies have shown that RIG-I like receptors can bind to host RNAs during lytic reactivation in PEL cells. Recent studies report that RIG-I like receptors can also bind to viral RNA to inhibit viral reactivation [14]. Formaldehyde RNA immunoprecipitation (fRIP-Seq) for RIG-I showed that KSHV RNAs including PAN, ORF50, ORF52, ORF57, ORF59, and vIL-6 interact with RIG-I to stimulate host innate immunity. To examine whether KSHV RNAs directly bind to DDX24 and DDX49, we performed RIPseq analysis on Rta-inducible TREx BCBL1 cells overexpressing DDX24 or DDX49, which showed enrichment of IE and E lytic KSHV transcripts after overexpression (Figure 6 and Table 1). These data suggest that DDX24 and DDX49 can interact with KSHV transcripts from within the nucleus to negatively regulate reactivation from latency. Some of the highest enriched transcripts that we have observed are K2, ORF70/K4/K4.1/K4.2, K5, and K9 which encode proteins involved in reactivation and immune modulation (Table 1). Of particular importance, K2 or vIL-6 is required for the survival of KSHV-infected B-cells. The K2 transcript has a short but structured 3′-UTR containing hairpins [9,64,65]. In a ribonomics experiment performed by the Damania lab, RIG-I was also found to bind to K2 KSHV RNA fragments in iSLK.219 cells via HITS-CLIP [17]. Additionally, although ORF58 and ORF59, which were enriched in both DDX24 and DDX49 RIP are not directly involved in reactivation, they are critical for viral replication. Several viral transcripts were exclusively enriched in the DDX49 pulldown, K8.1, K9, and ORF 65/66/67. The enrichment of late lytic transcripts ORF65 and K8.1 may explain the prolonged suppression of KSHV reactivation observed during the reactivation time course.

To address whether recognition of KSHV transcripts by DDX24 and DDX49 plays a role in the innate immune response, we tested whether type I interferon is induced in response to KSHV IE and E mRNAs, identified by our RIPseq data, in the absence of viral infection. Using HEK-Blue cells as IFN reporter cell lines, we observed that the protein expression level of type I interferon was induced by IE and E KSHV mRNAs, while mRNA level remains unelevated, demonstrating that KSHV IE and E transcripts can induce innate immune responses, probably via a post-transcriptional regulation mechanism.

More recently, we analyzed the RNAseq results from the input of our RIPseq analysis in TREx BCBL1 cells overexpressing either DDX24 or DDX49, compared to GFP-transduced cells. We identified a limited number of Interferon-responding genes whose expression levels were altered upon the overexpression of DDX24 and DDX49, including STAT2 and ADAR1, which can facilitate KSHV lytic reactivation [66]. While Ingenuity pathway analysis identified increased type I interferon response in DDX24 and DDX49 expressing cells, the extent of this induction was moderate. This is likely due to the fact that all other IFN inducing mechanisms are also triggered in induced TREx BCBL1 cells. To decipher the direct impact of DDX24 and DDX49 under these conditions will likely require generation of inducible expression systems, which is beyond the scope of this manuscript.

The race between host antiviral innate immunity and viral immune evasion strategies is an ongoing topic. Firstly, elucidating the recognition mechanism on DDX24 and DDX49 could help identify more RLRs sensing additional human viruses in the context of reactivation in the future. Studying RNA structure may reveal whether DDX24 and DDX49 have similar or divergent structures compared to RIG-I recognition motifs (PAMP motif). Moreover, the common feature of the sequence or secondary structure of KSHV transcripts could help develop new therapeutic strategies of KSHV infection and its associated disease. For instance, the inability of RIG-I to recognize unpaired exposed 5′ triphosphate (5′ppp) is used to develop novel therapeutic strategies. Secondly, identifying cellular RNA binding partners of DDX24 and DDX49 from the RNA IP dataset may lead to insight into cellular processes regulated by these helicases. snoRNAs are known to be binding partners of DEAD-box helicases, there may also be additional innate immune regulators and/or lncRNAs directly bound by these helicases in the nucleus. DDX24 competitively binds to FADD and RIP, preventing caspase 8 from activating NF-κB, which is necessary for KSHV reactivation and B cell survival [37]. The same apoptotic pathway is also targeted by multiple KSHV gene products [57] Therefore, cells may induce apoptosis in response to DDX24-dependent immune sensing of KSHV IE and E transcripts. Lastly, since DDX24 is targeted by microRNAs in both latently infected B cells, endothelial and fibroblast cells, it will be important to see whether KSHV-induced innate immune responses during reactivation from latency are different in the presence or absence of DDX24-targeting miRNAs.

## 5. Conclusions

In summary, based on our screen of human DEAD box helicases and the RIPseq analysis performed in Rta-inducible TREx BCBL1 cells, we identified two novel PRRs that negatively regulate KSHV replication in the context of reactivation from latency. DDX24 and DDX49 are mostly binding IE and E lytic KSHV RNAs, suggesting that DDX24 and DDX49 can recognize KSHV nucleic acids from within the nucleus, thereby inhibiting lytic reactivation. Understanding the molecular mechanisms by which DDX24 and DDX49 inhibit reactivation could lead to therapeutic strategies for EBV and KSHV malignancies whose pathogenesis is driven by both the latent and lytic phases of infection.

## Figures and Tables

**Figure 1 viruses-14-02083-f001:**
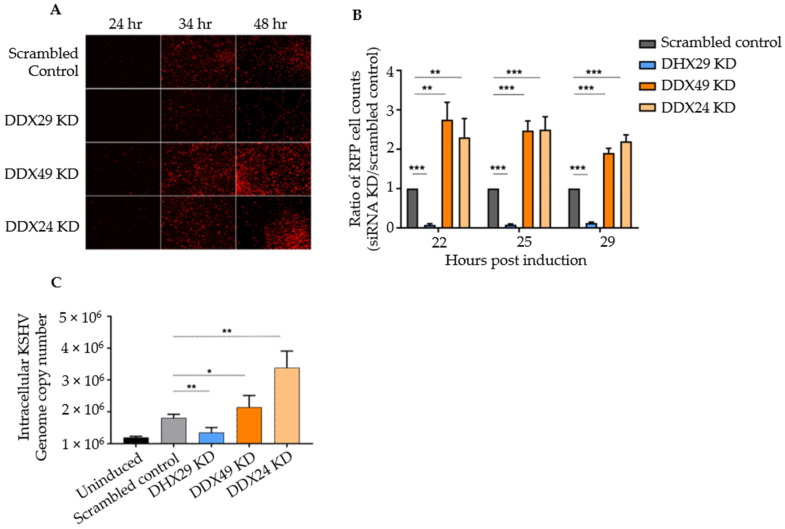
siRNA knockdown screen of selected DExD/H box helicases in Vero cells latently infected with rKSHV.219. (**A**) Fluorescence microscopy results of knockdown with scrambled siRNA control and with siRNAs against DHX29, DDX49, and DDX24 at 24, 34, and 48 h post-induction. All transient KD Vero cells looked healthy 28 h after siRNA transfection and prior to chemical induction, with negligible spontaneous background reactivation. (**B**) Counts of RFP^+^ cells for selected knockdowns with three random fields of vision per biological replicate. (**C**) Intracellular DNA copy number as determined by quantitative PCR under the same knockdown conditions. Values are expressed as a ratio normalized to the scrambled control (n = 4 independent trials, *** *p* < 0.001, ** *p* < 0.01, * *p* < 0.1).

**Figure 2 viruses-14-02083-f002:**
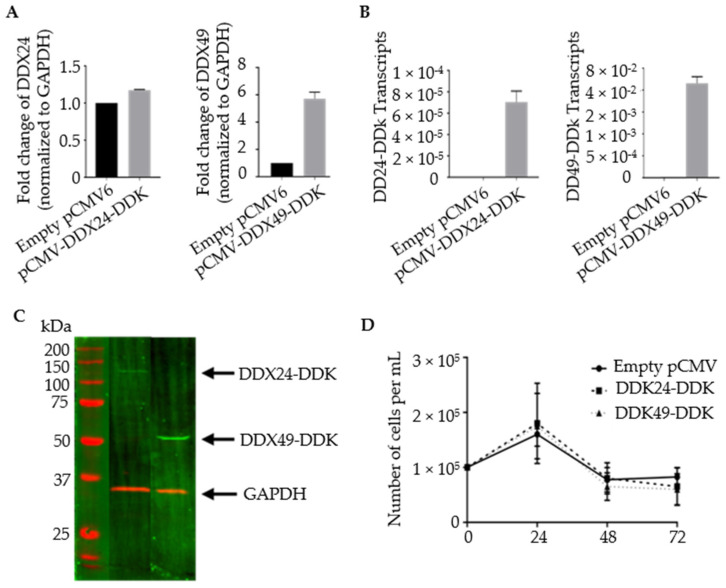
Validation of stably transfected BCBL-1 cells overexpressing DDX24-DDK and DDX49-DDK. (**A**) Relative transcript abundance of DDX24 and DDX49. Primers target gene body of endogenous DDX24 and DDX49. Steady state transcript levels include endogenous and DDK tagged helicases. (n = 1). (**B**) Relative transcript abundance of DDX24-DDK and DDX49-DDK templates. Primers target 3′ end of the helicase Open Reading Frame (ORF) and the DDK tag (n = 2). (**C**) Immunoblots of DDX24-DDK and DDX49-DDK utilizing anti-DDK antibody. (**D**) DDX24-DDK and DDX49-DDK stably transfected BCBL-1 cell viability during lytic reactivation of a 2 mM NaB induction time course (not significantly different from pCMV control).

**Figure 3 viruses-14-02083-f003:**
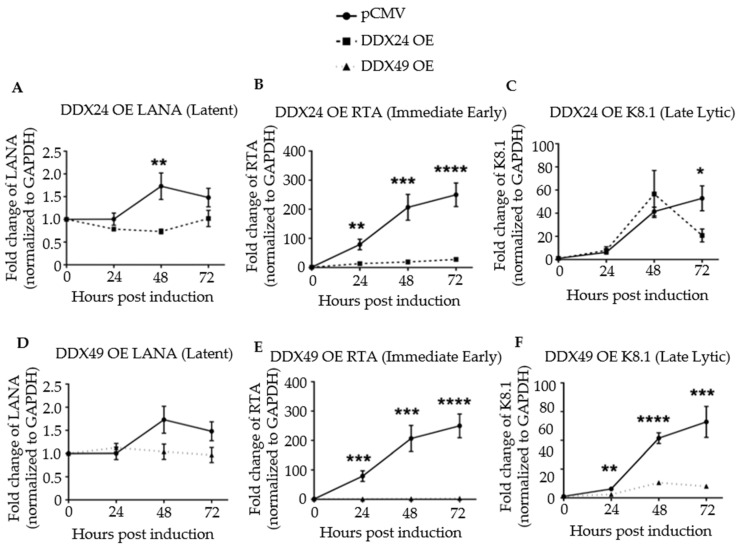
Expression of latent, immediate early, and late lytic genes in induced BCBL-1 cells overexpressing DDX24-DDK or DDX49-DDK. Cells were induced with 2 mM sodium butyrate and the levels of LANA, RTA and K8.1 transcripts determined by RT-qPCR at 24-, 48- and 72 h post-induction. (**A**) LANA transcript levels in BCBL-1 cells over-expressing (OE) DDX24-DDK or with empty pCMV vector. (**B**) RTA transcript levels in DDX24-DDK OE cells or pCMV cells. (**C**) K8.1 transcript levels in DDX24-DDK OE cells or with empty pCMV cells. (**D**) LANA transcript levels in DDX49-DDK OE cells or pCMV cells. (**E**) RTA transcript levels in DDX49-DDK OE cells or pCMV cells. (**F**) K8.1 transcript levels in DDX49-DDK OE cells or pCMV cells. Statistically significant difference was observed in mean gene expression of LANA and RTA between the empty pCMV vector and DDX24 (n = 3, LANA *p* = 0.02, RTA *p* = 0.0003, K8.1 not significant). There was a statistically significant difference in mean gene expression of Rta and K8.1 between the empty pCMV vector and DDX49 (n = 3, LANA not significant, Rta *p* < 0.0001, K8.1 *p* < 0.0001). * *p* < 0.1, ** *p* < 0.01, *** *p* < 0.001, **** *p* < 0.0001.

**Figure 4 viruses-14-02083-f004:**
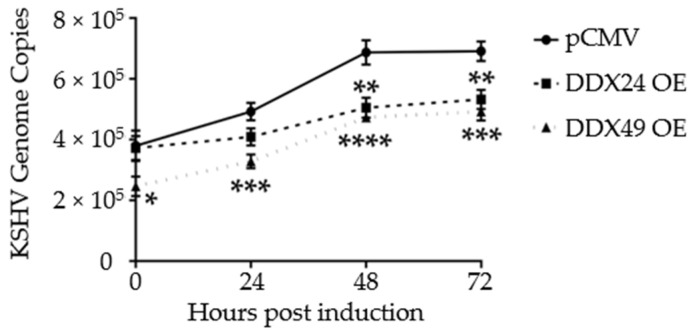
Intracellular KSHV genome copy number in BCBL-1 cells stably expressing DDX24-DDK, DDX49-DDK or empty pCMV vector during a 2 mM NaB induction time course. There was a statistically significant difference in mean genome copy number between the empty pCMV vector and DDX24 or DDX49 throughout the induction time course (n = 3, DDX24 *p* = 0.03, DDX49 *p* = 0.0002). (n = 3, * < 0.05, ** < 0.01, *** < 0.001, **** < 0.0001).

**Figure 5 viruses-14-02083-f005:**
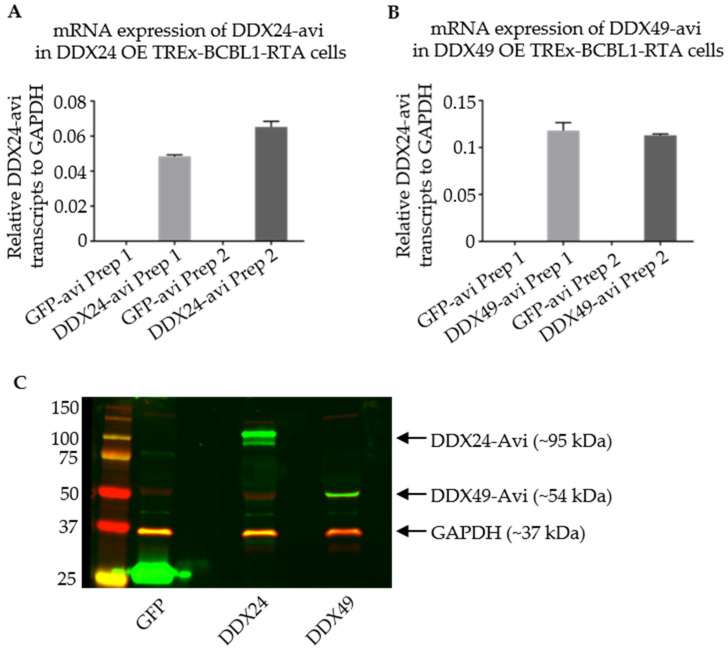
Validation of Stably Transduced TREx BCBL1-Rta cells overexpressing DDX24-Avi or DDX49-Avi for RNA IP. (**A**,**B**) qPCR representing relative DDX24-Avi and DDX49-Avi in TREx-BCBL1-Rta cells after infection with lentivirus from two individual preparations (Prep1 and Prep2) as labelled on *x*-axis. (**C**) Immunoblots utilizing anti-Avi antibody representing expression of DDX24-Avi and DDX49-Avi at their appropriate relative masses.

**Figure 6 viruses-14-02083-f006:**
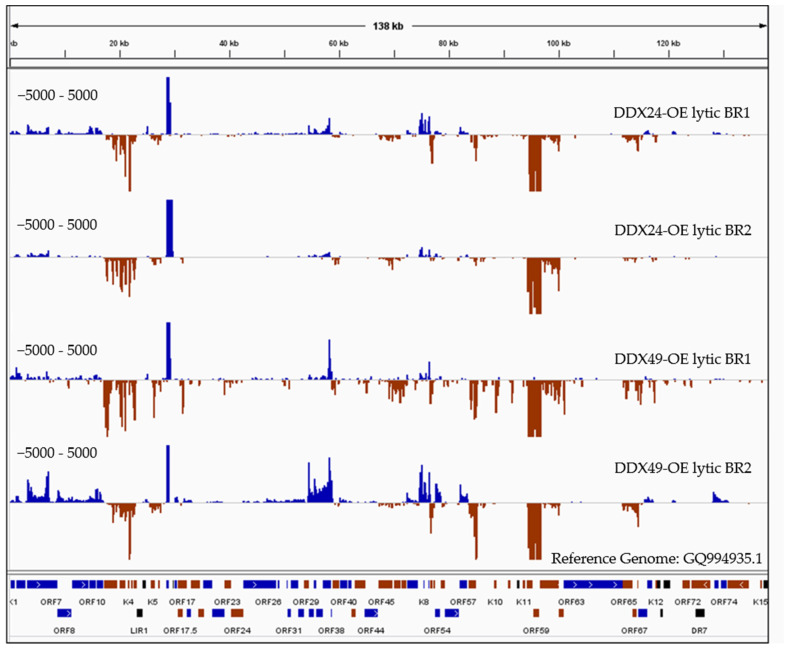
RNASeq coverage analysis of DDX24 and DDX49 associated KSHV transcripts compared to the control immunoprecipitated RNA upon Rta-induced reactivation. RNA samples immunoprecipitated using anti-Avi from DDX24/TREx BCBL1-Rta, DDX49/TREx BCBL1-Rta and control transfected TREx BCBL1-Rta cells were subjected to Illumina strand-specific Next Generation Sequencing analysis. The plot here represents the distribution of read counts per million (CPM) in DDX overexpressing samples normalized by subtracting the corresponding CPM values in control sample mapped along the annotated KSHV genome on the *x*-axis. The *y*-axis is showing the number of reads (coverage) on the positive or negative strand of the genome (blue and red).

**Figure 7 viruses-14-02083-f007:**
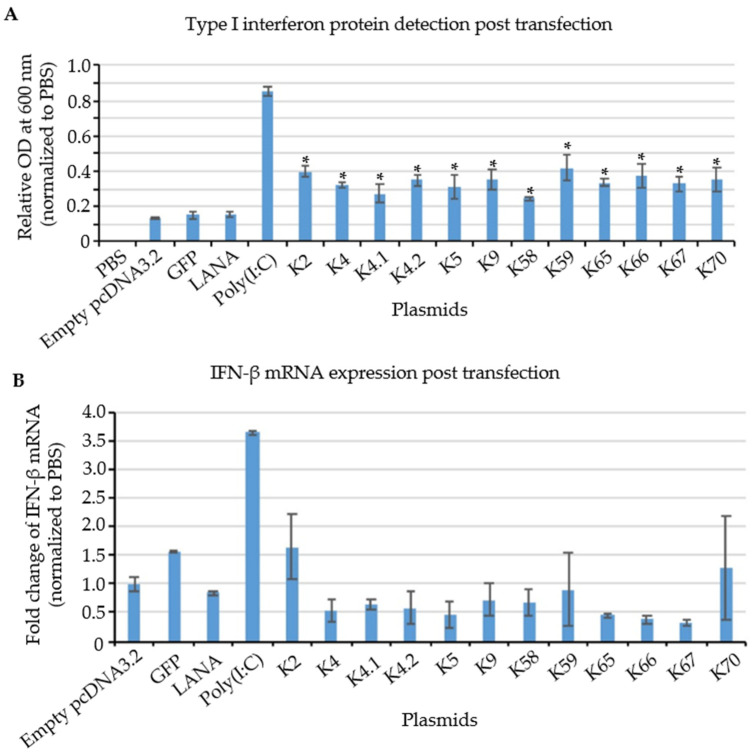
Relative expression of IFN protein and mRNA in KSHV IE and E gene transfected cells. (**A**) IFN protein detection by HEK-blue system. KSHV transcripts and positive control, poly(I:C) showed a significant increase in IFN protein expression, while negative controls, including LANA, GFP, and empty vector showed no change compared to PBS treatment. (n = 3, * *p* = 0.05) (**B**) RT-qPCR of IFN after 12 h post induction. No significant change was observed in the IFN mRNA among experimental and control groups. (n = 3).

**Table 1 viruses-14-02083-t001:** List of genes with highly enriched regions from RNA-IP analysis.

KSHV Gene	Phase	Expressed from Spliced mRNAs	KSHV Protein	Function	Reference
K2	Immediate-Early	Yes	viral interleukin-6 (v-IL6)	activates JAK/STAT, MAPK, and Akt signaling pathways to regulate B-cell proliferation and KSHV reactivation	[44,46]
ORF70/K4/K4.1/K4.2	Immediate-Early/Early	No	v-CCL3 (v-MIP-III), v-CCL3 (v-MIP-II)	ORF70: Govern the transition of the KSHV genome from latent to lytic phase. K4/K4.1: Homologs of cellular chemokines	[44]
K5	Immediate-Early	Yes	Modulator of Immune Recognition (MIR2)	encodes for a ubiquitin E3 ligase that can degrade MHC-I molecules	[45]
ORF58	Early	No	EBV BMRF2 homologue	A component of the tegument that interacts with the envelope	[47]
ORF59	Early	Yes	DNA polymerase processivity factor, Cytomegalovirus (CMV) and EBV homolog	A viral DNA polymerase processivity factor required for lytic DNA replication	[47]
K9 (only in DDX49)	Early	No	vIRF1	Homolog of cellular interferon regulatory factor	[48,49,50]
ORF65/66/67 (only in DDX49)	Early/Late	No	EBV BFRF2	Required for virion production	[51]

## Data Availability

The RNAseq data mentioned in this study can be found at the SRA database as BioProject: PRJNA862899 (RNA Immunoprecipitation of DDX24 and DDX49 in TREx-BCBL1-Rta).

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
