# Peer review of "DExD/H Box Helicases DDX24 and DDX49 Inhibit Reactivation of Kaposi’s Sarcoma Associated Herpesvirus by Interacting with Viral mRNAs"

_viruses, 2022, doi:10.3390/v14102083_

Round 1

Reviewer 1 Report

In their manuscript Serfecz and colleagues performed a knock-down screen of 22 DExD/H box helicases to identify potential involvement in the lytic reactivation of KSHV. Two proteins, DDX24 and DDX49, were found to carry antiviral activity. Overexpression of these proteins reduced lytic viral gene expression and viral DNA replication. The authors further show, by using RNA immunoprecipitation, binding of DDX24 and DDX49 to viral transcripts. In line, ectopic expression of KSHV genes whose transcripts were identified as being bound to DDX24 and DDX49 activated IFNα/β secretion and IFNβ mRNA expression. Thus, these findings reveal two new cellular proteins that could function as PRRs during lytic reactivation of KSHV towards inhibition of the lytic cycle.

The introduction is clearly written, detailed and contains sufficient background that helps understand the study performed and the results obtained. The results are well described and the discussion is comprehensive and detailed. The authors findings are supported by the data presented.

Specific comments:

1.     The list of DExD/H box helicases that were tested by using siRNA screen is included in the supplementary material. Based on the initial screen the authors further examine the effects of DHX29, DDX24 and DDX49. It is not clearly stated that the other 19 tested proteins did not affect lytic reactivation and the authors may consider the inclusion of the entire screen results and indicate its limitation.

2.     Figure 2B – Measurements of the transcripts are not clear as cells transfected with a control empty plasmid are not expected to show any product by RT-PCR.

3.     Figure 6 shows the association of DDX24 and DDX49 with KSHV transcripts. Has binding of these proteins to cellular transcripts during lytic reactivation been detected?

4.     Figure 7 – there is no indication of the number of biological repeats. Furthermore, the authors describe significant changes yet no statistical data are shown.

5.     The authors indicate that DDX24 and DDX49 detect the KSHV mRNAs in the nucleus however no data is provided to support this suggestion.

Minor comments:

1.     Page 1, line 40 –angiogenesis and the appearance of “dark red lesions” represent the characteristics of KS lesions yet are not the outcome of transformation.

2.     Page 2, line 47 – PEL is mostly diagnosed in AIDS patients however it may develop in HIV-negative individuals as well.

3.     Page 6, line 285 – PAN promoter is an early lytic viral gene instead of an immediate-early.

4.     Page 6, line 287 – the description of the experiment is not clear and only 29 hr induction is indicated although more time points are shown in Fig. 1.

5.     Figure 1 – the figure does not show the results of the screen and therefore its title needs to be modified.

6.     Figure 1 – the authors show images of the cells that were taken 24, 34 and 48 hr post induction but then count RFP-positive cells at different time points. Can this be explained?

7.     Figure 2B, left – transfected DNA is mislabeled and should by DDX24.

8.     Page 11, line 434 – “KSHV reactivation from within the nucleus” – this statement is unclear. Please explain.  

9.     Page 15, line 537 – the requirement of nuclear localization for KSHV mRNA export is not clear.

Reviewer 2 Report

In the manuscript entitled “ DExD/H box helicases DDX24 and DDX49 inhibit reactivation 2 of Kaposi’s Sarcoma associated herpesvirus by interacting with 3 viral mRNAs “ the authors screened DEAD box helicases and identified two (DDX24 and DDX49) that recognizes KSHV nucleic acids and inhibit the lytic reactivation. The authors concluded that this DDX24 and DDX49 could be a potential therapeutic target.

There are few concerns that needs to be addressed

1. The authors used 22 siRNA against different DexD/H box helicases. The authors should check the KD efficiency either by WB or atleast by real time PCR. The siRNAs may be validated but the authors should make it clear that the siRNAs are working in their system.

2. In Fig. 2D, the authors have shown that over-expression  of DDX24 and DDX49 did not have significant change in viability as compared to that of the vector control post 72 hrs of reactivation. This data is confusing because if BCBL1 cells (with vector control) are reactivated with BA and TPA for 72h there will be a considerable reactivation and that should affect the cell viability. In the result the authors have shown that the vector control has more viable cells compared to the DDX24 and DDX49. Can the authors please explain this.

3. The authors have shown that KD of the DHX29 demonstrated 90% reduction in reactivation and the viral copy number decreased to 2/5. This is a very significant change and the authors have also mentioned that “DHX29 acts as a pro-viral sensor”. But the authors then did not follow up with DHX29 because they didn’t observe any change in the viral gene expression post KD. The authors must provide ample evidence to support the fact of dropping the DHX29 and not following it up.

4. In fig3 the authors have demonstrated the effect of KD of DDX24 and DDX49 on the expression of LANA, RTA and K8.1. As evident from the results, the changes observed in RTA is very significant. For RTA and for K8.1 the 0h expression is shown as 0. Can the authors clarify how they analysed the data if the 0 h foldchange is 0. What is control sample based on which they did this?

Moreover DDX40 OE was found to totally shutdown the RTA expression. What can be the possible explanation for this.

5. OE of DDX24 post 48 h infection has a higher expression of k8.1 (late lytic gene) whereas the early lytic gene expression is significantly less. Authors should explain this findings.

6. In fig.4 the authors have shown that the DDX49 OE cells have a considerably less viral copy numbers compared to the DDX24 OE or the vector control. What can be the possible reason behind this?

The authors should address these queries and in some places in the text proper reference is missing.

Reviewer 3 Report

It has been previously described that RLR pathway restricts KSHV lytic reactivation (Nat Commun. 2018 Nov 19;9(1):4841.), while ADAR1 facilitates KSHV lytic reactivation (Cell Rep. 2020 Apr 28;31(4):107564.). In this manuscript, the authors report that other DExD/H box helicases DDX24 and DDX49 also inhibit KSHV lytic reactivation by interacting with viral mRNAs. Overall, it can be a good addition to the field. However, several issues should be addressed as described below.

Major Issues

1. In Figure 1, please add GFP signal image in panel A. Please modify labeling like T24, T22 in panel A-B and 24 h, 22 h would be acceptable. Please describe the methods related to panel B. Gene expression detection after knockdown using qPCR and/or western blot should be included in Figure 1. Please detect viral lytic RNA expression level after knockdown. Also, ratio of RFP positive cell from 22 DExD/H box helicases should be added in Table S1.

2. Result 3.2 should be included in 3.1 and put the corresponding result in supplemental information.

3. move result 3.3 in supplemental information.

4. Please detect intracellular DNA copy number after DDXs overexpression.

5. move result 3.5 in supplemental information.

6. In Figure 7, the experimental design is not appropriate. To test their ability of viral transcripts identified from our RIPseq data to induce a type I interferon response, transfect cells with RNA bound to DDXs and detect IFNβ using ELISA kit and qPCR. Alternatively, transfect cells with in vitro transcribed viral RNA.

Minor Issues

Please check English editing and style, such as not italicize in line 142-147. Please use uppercase letter in each panel same as in the main text.

Round 2

Reviewer 2 Report

The authors responded to all the queries and explained them in the manuscript. 

Author Response

We checked the spelling in the manuscript and resubmit the revised version. No further comments to address.

Reviewer 3 Report

After first revision from the authors, there are still some issues to be concerned.

1. In line 284-287, it says “The Vero cell line is latently infected with a recombinant virus, 284 rKSHV.219 which expresses the red fluorescent protein (RFP) from the KSHV early lytic 285 PAN promoter, and the green fluorescent protein (GFP) from the EF-1α promoter, and 286 with the gene for puromycin resistance as selectable marker.” Does GFP signal represent the cells without KSHV reactivation?  Is it more accurate to calculate the RFP/GFP ratio?

2. Although the authors claimed that 5 commercial siRNAs per gene were used, it is better to determine the knockdown efficiency for research accuracy and rigor.

3. Total 22 DExD/H box helicases were tested using siRNA knockdown for their effects on KSHV reactivation. Although the other 19 DExD/H box helicases did not show a significant change, the corresponding results should be added.
